# Brief communication "3D reconstruction of a collapsed rock pillar from web-retrieved images and terrestrial LiDAR data – The 2005 event of the West face of the Drus (Mont-Blanc massif)"

Antoine Guerin[1], Antonio Abellán[2], Battista Matasci[3], Michel Jaboyedoff[1], Marc-Henri Derron[1] and Ludovic Ravanel[4]

[1]Risk Analysis Group, Institute of Earth Sciences, University of Lausanne, Switzerland
[2]Scott Polar Research Institute, University of Cambridge, United Kingdom
[3]Bureau d'Etudes Géologiques SA, Aproz, Switzerland
[4]EDYTEM, University Savoie Mont Blanc – CNRS, Le Bourget du Lac, France

*Correspondence to*: Antoine Guerin (antoine.guerin@unil.ch)

**Abstract.** In June 2005, a series of major rockfall events completely wiped out the Bonatti Pillar located in the legendary Drus West face (Mont-Blanc massif, France). Terrestrial LiDAR scans of the West face were acquired after this event but no pre-event point cloud is available. Thus, in order to reconstruct the volume and the shape of the collapsed blocks, a 3D model has been built using photogrammetry (SfM) based on 30 pictures collected on the Web. All these pictures were taken between September 2003 and May 2005. We then reconstructed the shape and volume of the fallen compartment by comparing the SfM model with terrestrial LiDAR data acquired in October 2005 and November 2011. The volume is calculated to 292'680 m$^3$ (± 5.6 %). This result is close to the value previously assessed by Ravanel and Deline (2008) for this same rock-avalanche (265'000 ± 10'000 m$^3$). The difference between these two estimations can be explained by the rounded shape of the volume determined by photogrammetry, which may lead to a volume overestimation. However it is not excluded that the volume calculated by Ravanel and Deline (2008) is slightly underestimated, the thickness of the blocks having been assessed manually from historical photographs.

## 1 Introduction

The Drus (3'754 m a.s.l.) are emblematic summits of the Chamonix valley situated in the Mont-Blanc massif (France). Since the middle of last century, the Petit Dru West face (1000 m-high, 3'730 m a.s.l.) is affected by intense erosion which has significantly modified the morphology of this peak (Ravanel and Deline, 2006 and 2008; Fort et al., 2009). In June 2005, a rock pillar (the Bonatti Pillar) estimated to be around 265'000 ± 10'000 m$^3$ by Ravanel and Deline (2008) collapsed, destroying forever numerous climbing routes. The assessment of this volume by Ravanel and Deline (2008) was performed in two steps: (a) identification on photos of different rock elements (slabs, dihedrons, overhangs) whose dimensions (height, width, depth) can be compared with compartments now collapsed; and (b) measurements of these dimensions on terrestrial LiDAR scans acquired just after the event in October 2005. Historical photographs of the West face taken from different

viewpoints facilitate the estimation of the thickness of the missing elements, which remains the most difficult dimension to determine. Under this method, the assessment of rock thickness (8 meters on average) represents the greatest source of uncertainty since the height and width of the rock-avalanche scar could be very accurately measured on the October 2005 LiDAR data. Note that these LiDAR scans correspond to the oldest reference and no 3D model is available before the major

event of June 2005. Thus, in order to get the pre-event topography of the Petit Dru West face, we collected several pictures dating from 2003 to 2005 from different web picture hosting services and a 3D photogrammetric model was reconstructed. Such an approach was already used in different research areas such as cultural heritage conservation: precursor of this "crowdsourced" technics, Grün et al. (2004 and 2005) reproduced in 3D the statue of the Great Buddha of Bamiyan (Afghanistan) using a series of pictures obtained from the Internet. More recently, many historians, archaeologists or

architects (e.g. Furukawa et al., 2010; Doulamis et al., 2013; Ioannides et al., 2013; Kyriakaki et al., 2014; Santos et al., 2014) took advantage of the large amount of images available online to preserve and keep a digital record of cultural and historical heritage using Structure-from-Motion (SfM) algorithms (Snavely et al., 2008). According to the *New York Times* (Estrin J., 2012), over 380 million pictures are uploaded on Facebook every day and other authors such as Stathopoulou et al. (2015) or Vincent et al. (2015) have used crowdsourced imagery to virtually replicate heritage objects destroyed by natural

disasters, armed conflict or terrorism. Examples include the stone bridge of Plaka (Greece), the city of Kathmandu before and after the 2015 Earthquake and several artworks at the Mosul Museum (Iraq).

In geosciences, conventional photogrammetry has long been used for Digital Elevation Model (DEM) generation but it is only recently that SfM has popularized the use of 3D point clouds in this field (e.g. Firpo et al., 2011; Salvini et al., 2013;

James and Robson, 2014; Lucieer et al., 2014). The review conducted by Eltner et al., (2016) shows that the annual number of publications that refer to SfM has really exploded since 2014, particularly in the fields of soil erosion, glaciology and fluvial morphology. This method is surprisingly straightforward to implement and also relatively accurate when compared to other techniques such as ground-based LiDAR data. In 2013, Fonstad et al. obtained differences of about 0.1 m (in X, Y and Z) between these two methods. In addition, new technologies such as Unmanned Aerial Vehicles (UAV) combined with

SfM have modernized and revolutionized investigations on several Earth surface phenomena (Abellán et al., 2016; Smith et al., 2016). For instance, Turner et al. (2012) and Lucieer et al. (2014) obtained 4 cm errors comparing DEM from UAV-SfM to differential Global Positioning System (dGPS) ground control points. In 2016, Bakker and Lane have innovated by showing the potential to couple archival aerial photographs and SfM algorithms to quantify morphological changes in a river-floodplain system at a decadal scale. However, despite all these recent advances, paleotopographic reconstruction based

on old terrestrial images or orthophotos has been rarely used in the field of geohazards to improve erosion rate quantification (Oikonomidis et al., 2016). For this reason, the aim of this Short Note is to illustrate the potential to merge ground-based LiDAR measurements with terrestrial SfM point clouds made from publicly available images. This allows traveling back in time in order to better quantify past natural disasters. More specifically, this Short Note reports the results of the 3D reconstruction of the Drus West face before the Bonatti Pillar collapse in June 2005.

## 1.1 Geological and structural setting

From a geological point of view, the Mont-Blanc crystalline range describes a broad ellipse elongated in the NE-SW direction extending from the Val Ferret (Valais, Switzerland) to the Chapieux Valley (Savoie, France) (Fig. 1A). The central part of the massif develops on the Aosta Valley (Italy) and Haute-Savoie (France) and it consists of two major petrological units: plutonic rocks (granites), mainly, and metamorphic rocks (gneiss and micaschists) which merge near the summit of Mont Blanc (Fig. 1A and 1B). From Southwest to Northeast, granites also pass of an intrusive position in gneiss to a tectonic contact materialized by the *faille de l'Angle* ("de l'Angle fault", Fig. 1B) (Epard, 1990). This fault separates the Mont-Blanc massif in two sections: an internal part essentially granitic and a more metamorphic external part (Epard, 1990; Steck et al., 2000 and 2001). The Petit Dru West face presents a coarse-grained calk-alkaline granite, which was formed during the Hercynian orogeny and dated from 305 ± 2 million years (Bussy et al., 1989; von Raumer and Bussy, 2004; Egli and Mancktelow, 2013). The steep rock cliff (average dip angle of 75°) is cut by a set of two large sub-vertical fractures oriented 238°/85° and 303°/79° which form wedges and by four other joint sets (especially 106°/33°) which form deep overhangs (Ravanel and Deline, 2008; Matasci et al., 2015). These very persistent dihedral structures (mean trace length of 80 m) promote the collapse of large compartments and have played a major role (Matasci et al., 2015) during the large rockfall events of summer 2005 and fall 2011 (Fig. 1C).

<< FIGURE 1 >>

## 2 Material and methods

The 3D reconstruction of the Drus West face was carried out using 30 web-retrieved images from different picture hosting services (*Flickr.com*, *SummitPost.org* and *Camptocamp.org*, see Appendix A) and a commercial photogrammetric software (Agisoft PhotoScan, 2014 – version 1.0.3). The georeferencing/alignment procedure of point clouds was done with CloudCompare (Girardeau-Montaut, 2015) software (version 2.7.0) and an estimation of the missing volume was then performed on 3DReshaper (Technodigit, 2014) software (2014 MR1 version) by comparing the SfM point cloud with terrestrial LiDAR scans acquired after the event.

### 2.1 Selection of photographs from Internet

Before the June 2005 rock-avalanche, the Drus West face was affected by major rockfalls in September 1997 (27'500 ± 2'500 m$^3$) and August 2003 (6'500 ± 500 m$^3$) (Ravanel and Deline, 2008). These events have significantly modified the morphology of the pillar between 3'160 m a.s.l. and 3'460 m a.s.l. (Fig. 1D) and we thus looked for photographs taken between September 2003 and May 2005. This was carried out by looking at the Exif metadata which are publicly available within the three above-mentioned imagery repositories. After a visual checking, 30 pictures taken from different viewpoints and with a mean size of 500 Ko were selected (Fig. 2 and Appendix A). Note that due to a limited number of available

images, we were forced to choose pictures taken in different seasons. However, in winter, snow is hardly present in the steep Drus faces and except at the foot of the cliff, there is no snow in the area of interest of the Bonatti Pillar on the 30 selected images (Fig. 2).

## 2.2 Ground-based LiDAR data acquisition

In order to obtain a 3D model of the entire Drus West face with a high and homogeneous density of points ($\sim$250 points/m$^2$, i.e.1 point every 6.2 cm), we merged the LiDAR scans from two different measurement campaigns carried out in October 2005 and November 2011. The 2005 point cloud (assembly of 3 scans, 24 million points) represent only the upper part of the

face and was acquired from the Flammes de Pierre ridge (Fig. 2) with a medium-range laser scanner (Optech ILRIS-3D) (Ravanel and Deline, 2006). The 2011 point cloud (assembly of 3 scans, 24 million points also) of the whole face has been acquired with a long-range laser scanner (Optech ILRIS-LR) from the right lateral moraine of the Drus glacier, situated around 2'500 m a.s.l. (Fig. 2).

## 2.3 Georeferencing and alignment of LiDAR scans

In the absence of a fairly accurate DEM (the resolution of the IGN's DEM is only 30 m in this sector), both datasets were georeferenced using the scanner position measured by dGPS, then aligned with respect to the vertical axis using the coordinates of several points distributed in the cliff and measured with a total station. The scans were then aligned with each other using Iterative Closest Point (ICP) algorithms (Besl and McKay, 1992) but only applied to stable parts (manually selected because of the different viewpoints) because between these two acquisitions, two major rockfalls occurred in

September 2011 (4'530 $\pm$ 200 m$^3$) and October 2011 (54'730 $\pm$ 400 m$^3$) in the June 2005 rock-avalanche scar area (Fig. 6D). These volumes were determined by comparing the 2005 and 2011 LiDAR acquisitions and include the "small" rockfalls (range of volumes: 1 m$^3$ to 426 m$^3$) detected between October 2005 and September 2008, by an annual LiDAR monitoring carried out from the Flammes de Pierre ridge (Ravanel, 2010). The points belonging to all these rockfall events have therefore been removed from the merged cloud and a volume of 59'260 m$^3$ is to be subtracted from the estimated volume for

the Bonatti Pillar collapse, given by the result of the comparison between the pre (SfM model) and post-event (2005/2011 merged and "cleaned" LiDAR cloud).

## 2.4 Construction and alignment of the SfM point cloud

The workflow described by Smith et al. (2016) was used to construct a point cloud of the former Drus West face with Agisoft PhotoScan. All selected pictures were aligned during this procedure and the final model (Fig. 2) that represents the

north-western side of the Aiguille Verte and Drus (Nant Blanc catchment) consists of 895'300 points, with a mean density of 0.42 points/m$^2$ (Fig. 3D). Note that in the Bonatti Pillar sector (the area of interest), this value is slightly higher and reaches a

medium value of 0.65 points/m$^2$ and no ground control points have been imposed when generating the 3D model on Agisoft PhotoScan. The SfM point cloud was then roughly scaled and aligned on the 2005/2011 merged LiDAR point cloud by selecting several equivalent point pairs (a dozen) sufficiently distant from each other. After this, the SfM model was cut into thirty parts with an octree structure in order to accurately align and scale each portion independently on the LiDAR point cloud. As highlighted by Wujanz et al. (2016), ICP algorithms (Besl and McKay, 1992) were only applied to stable parts so as not to bias the comparison values detected in the Bonatti Pillar area. Furthermore, aligning and scaling each part independently compensates for the fact that no ground control points have been imposed in Agisoft PhotoScan. This procedure makes it possible to gradually deform the SfM cloud and to optimally adjust each section on the reference LiDAR point cloud. However, the overall shape of the SfM cloud is very wavy (Fig. 3D and 3F) and because of this, the average deviation in the stable areas reaches ± 1.17 m (Fig. 3E).

<< FIGURE 3 >>

## 2.5 SfM/LiDAR comparison and rockfall extraction

The first step to perform a point-to-mesh comparison was to transform the 2005/2011 merged LiDAR point cloud into a reference triangular mesh. All the points were used for the mesh generation and a maximum length of triangle edge of 5 m was set to fill the existing holes in the point clouds (zones masked by the relief). Unlike the point-to-point comparison, the point-to-mesh comparison calculates the orthogonal distance between both entities, which corresponds to the shortest distance between a point and the nearest triangle. Figure 3A shows the result of this comparison but also the points (in red on the Fig. 3B) that were extracted from the SfM cloud and associated with the Bonatti Pillar collapse. The point extraction was carried out on the basis of the method defined by Tonini and Abellán (2014). This method is illustrated in Fig. 4 and includes four steps: (a) Definition of a Level of Detection (LoD ± 1.2 m in our case, in agreement with the average deviation observed in the stable areas) and three-color distribution of comparison values: red for positive deviations, green for the points between ± 1.2 m and blue for negative deviations (Fig. 4A), (b) Color filtering to keep only the red points associated with positive deviations and in which the points associated with the Bonatti Pillar collapse are present (Fig. 4B and 4C), (c) Noise reduction using the Nearest Neighbor Clutter Removal algorithm (Byers and Raftery, 1998) which is based on the spatial density of points in 3D and (d) Individualization of rockfalls with the DBSCAN algorithm (Ester et al., 1996) which uses a distance criterion to explode a cloud into a sub-group of clouds (Fig. 4D).

<< FIGURE 4 >>

## 2.6 Volume calculation

We estimated the June 2005 rock-avalanche volume by constructing a closed mesh. For this purpose, the points extracted from the SfM cloud were first converted into a triangular mesh (Fig. 5A to 5D) to generate a surface whose contour (the free

border) has been extracted automatically (red contour in Fig. 5). However, unlike the LiDAR mesh, all the points of the extracted SfM cloud have not been preserved for the generation of this second mesh. Indeed, we decided to subsample the SfM cloud and retain only 1 point out of 10 (Fig. 5B and 5C), then to smooth the mesh obtained (Fig. 5D and 6D) in order to limit as far as possible the undulation effect highlighted in section 2.4. This smoothing procedure is accompanied by an interpolation of new points and was first tested on the profile P2 located within the stable area framed in Fig. 3. Figure 6 shows that the smoothing makes it possible to generate a substantially less undulating profile and thus much closer to the LiDAR profile. The dispersion diagram of Fig. 6C illustrates this aspect since the smoothing allows to minimize the large deviations and to significantly reduce the average deviation by ± 0.76 m. By applying this correction factor, the average deviation therefore changes from ± 1.17 m to ± 0.41 m in stable areas, final value used to define the uncertainty on the depth of the estimated volume.

<< FIGURE 5 >>

The red contour (3D polyline) of the smoothed mesh was then orthogonally projected onto the reference LiDAR mesh (Fig. 5E) in order to divide it into two parts and keep only the triangles located inside the projected contour (delimitation of the rockfall scar, Fig. 5F and 5G). The gap between both contours was filled by a third mesh, which corresponds to the thickness of the fallen volume (Fig. 5H). Finally, we merged these three surfaces to generate a closed mesh (Fig. 5I). The volume of the rockfall event is then given by the sum of the tetrahedrons volumes forming the closed mesh. In addition, in order to assess another error on the volume calculation but only related this time to the SfM method itself, we created two other SfM models by importing respectively 84 % and 67 % of the pictures used to construct the first point cloud.

<< FIGURE 6 >>

## 3 Results and discussion

The comparison between the SfM point cloud and the LiDAR mesh of 2005/2011 gives a volume of 351'940 $m^3$ (Fig. 7A and 7B). As specified in section 2.3, this volume includes the rockfall events that occurred in September and October 2011 and we had to subtract 59'260 $m^3$ (Fig. 7D) from this value to properly assess the June 2005 rock-avalanche volume. Therefore, the final value is equal to 292'680 $m^3$, which is quite close to the 265'000 ± 10'000 $m^3$ (i.e. ± 3.8 %) estimated by Ravanel and Deline (2008) since the uncertainty on the thickness of the estimated volume (± 0.41 m) which arises from all steps of the data processing (scaling and alignment by parts, SfM point cloud subsample and mesh smoothing) gives an error range equal to ± 16'400 $m^3$ (i.e. ± 5.6 %). Furthermore, the volumes estimated with the two other SfM models are respectively equal to 311'970 $m^3$ and 326'240 $m^3$. Thus, if we consider the volume of 292'680 $m^3$ as the most reliable estimation, the relative error between the three SfM models and related only to the SfM method itself, is respectively equal

to ± 6.6 % and ± 11.5 %. Given the large difference of density of points observed between the SfM model and the LiDAR point cloud (about 500 times higher for the LiDAR), this uncertainty value is acceptable and consistent (same order of magnitude) with the one linked to the whole data processing and the one given by Ravanel and Deline (2008).

<< FIGURE 7 >>

The lower density of points (0.65 points/m$^2$) of the SfM cloud is also found in the overall shape of the calculated volume, which is quite rounded (Fig. 7A) and lacks morphological details such as overhangs visible in the upper part of the Bonatti Pillar (Fig. 1D). This lack of details is due to the medium resolution of the images that we used to generate the SfM model,
the fact that most of the photographs were taken far from the face (Fig. 2) but also to the smoothing procedure which has clearly rounded the corners and edges of the SfM mesh. However, this step was necessary to minimize the undulation effect observed in the SfM point cloud and thus, significantly reduce the uncertainty on the depth of the estimated volume. Note that without this smoothing stage, the volume estimated for the Bonatti Pillar collapse reaches a value of 353'800 m$^3$ which is characterized by a relative error of ± 20.9 %, only related to the scaling/alignment procedure and the fact that no ground
control points were imposed during the SfM point cloud generation. On the other side, the rounded shape of the volume determined from the coupling SfM/LiDAR suggests that the 292'680 m$^3$ (± 5.6 %) calculated could be overestimated. The results shown in Fig. 3A and 3B head in this direction since the large positive deviations observed inside the white ellipses do not correspond to rockfall events (verified on pictures), but artefacts that form "tips" in the SfM point cloud. These "tips" are clearly visible on the longitudinal profile that passes through the LiDAR and SfM point clouds in Fig. 3C and also
present within the raw points assigned to the Bonatti Pillar collapse (Fig. 5A and 6D). Fortunately, the smoothing of the raw and subsample SfM mesh enabled to minimize these large deviations. These local deformations are certainly linked to the fact that the selected images were taken in different seasons, with different lighting-shading conditions, with different cameras and their resolution is quite variable (Fig. 2).

In contrast, we could reproduce accurately the lateral boundaries of the collapsed volume as well as the height of the Bonatti Pillar. Figure 7C perfectly illustrates this aspect since the June 2005 rock-avalanche volume exceeds only in one place (at the top left) the scar limits (white dashed line) defined by Fort et al. (2009). Besides, this difference was expected because this area corresponds to the upper left part of the October 2011 rockfall event (Fig. 7D). However in this work, we were not looking for a highly accurate volume but to assess the potential of merging ground-based LiDAR acquisitions with terrestrial
SfM made from web-retrieved images for quantifying past natural disasters. With this in mind, it was possible to define a range of relative error for the volume calculation according to the number of pictures used to generate the SfM model: 10.4 % in the case of 30 pictures (difference of 27'680 m$^3$ between the volume of 292'680 m$^3$ and the reference value of 265'000 m$^3$, chosen because of its lower uncertainty: ± 3.8 %) and 23.1 % with 20 pictures (difference of 61'240 m$^3$ compared to 265'000 m$^3$). This suggests that the accuracy of the volume could be improved if more than 30 images would have been

available. Note that these error percentages could have been higher if the LoD chosen (± 1.2 m in our case) was lower (e.g. ± 1 m) since more points would have been extracted from the comparison and associated with the volume of the June 2005 rock-avalanche. Nevertheless, it is not excluded that the volume determined by Ravanel and Deline in 2008 is slightly underestimated because even if accurate measurements were performed on the LiDAR mesh of October 2005, there is no 3D

model available before the collapse. For such volume ranges, it is often the thickness that is difficult to correctly assess and a small variation (e.g. 15 cm) can modify the final result of several thousands of $m^3$. In the specific case of the Bonatti Pillar (500 m high for 80 m wide), a depth variation of 1 m could change the final volume of about 40'000 $m^3$. Finally, it is important to specify that both volumes fallen in September and October 2011 play a significant role in our estimations. However, given the uncertainties mentioned in section 2.3 – the volumes were calculated by comparing the October 2005

LiDAR point cloud of the Flammes de Pierre to the November 2011 LiDAR triangular mesh of the Drus glacier – the values are pretty accurate and it is not these estimations that most influence the final result.

## 4 Conclusion

The method described in this Short Note has worked remarkably well for the Petit Dru West face, which is a legendary peak photographed since decades and from several corners of the Chamonix Mont-Blanc Valley. However, it is important to

highlight that the same method would have been difficult to implement on a less well-known site, where fewer images could have been collected and downloaded from picture hosting services on the World Wide Web. Another issue may be the limited number of viewpoints that exist on a study site because it is necessary to turn around the area of interest to create a good quality SfM model. In the field of natural hazards, digitizing of old photographs coupled to SfM methods is to be taken into account because it can deliver extremely useful data on the morphologies of the past. In some cases, this could allow to

go back to the beginning of the last century and even in 1860 for the Drus with the different photographs (daguerreotypes) of the Bisson brothers, two pioneers of the French photography.

## Acknowledgements

The authors would like to acknowledge the Swiss National Science Foundation (SNSF, grants 200020_146426 and 200020_159221) for supporting this research. Second author was granted with a Marie Curie fellowship (Project ref.:

705215). In addition, we would like to thank the authors of the pictures extracted from *Flickr*, *SummitPost* and *CamptoCamp* (see Appendix A for a detailed description) but also the support of the Chamonix Mont-Blanc Helicopters (CMBH) company for reaching the stations from which terrestrial LiDAR acquisitions were performed. Finally, special thanks go to Prof. Jean-Luc Epard for his explanations concerning the interpretation of the geotectonic setting of Mont-Banc massif.

**Appendix A: Photo credits**

Links to the 30 web-retrieved images that were downloaded from the following websites: *Flickr.com*, *SummitPost.org* and *Camptocamp.org*.

*Flickr.com (15)*

https://www.flickr.com/photos/42624864@N08/5765604229

https://www.flickr.com/photos/markhorrell/17225632811

https://www.flickr.com/photos/phileole/520418709

https://www.flickr.com/photos/phileole/520390144

https://www.flickr.com/photos/phileole/520419341

https://www.flickr.com/photos/mvcchris/9697023856

https://www.flickr.com/photos/robonabike/4568776704

https://www.flickr.com/photos/francoisdorothe/5451738425

https://www.flickr.com/photos/29922628@N08/3192264930

https://www.flickr.com/photos/davduf/1075398

https://www.flickr.com/photos/bengalshare/952842570

https://www.flickr.com/photos/tsa-climbing/6505792537

https://www.flickr.com/photos/ebbandflow/4500495087

https://www.flickr.com/photos/ebbandflow/4501086770

https://www.flickr.com/photos/jd-davis/15930404616

*SummitPost.org (12)*

http://www.summitpost.org/the-dru-as-seen-from-the-gran/40929/c-150757

http://www.summitpost.org/aiguille-verte/84226/c-183839

http://www.summitpost.org/aiguille-verte/84227/c-183839

http://www.summitpost.org/at-sunset-in-winter/85906/c-150757

http://www.summitpost.org/les-drus-from-mere-de-glace/116269/c-150757

http://www.summitpost.org/aiguille-du-dru/112906/c-182555

http://www.summitpost.org/aiguille-du-dru/112230/c-182555

http://www.summitpost.org/aiguille-verte/112911/c-182555

http://www.summitpost.org/aiguille-du-dru-flammes-de-pierre/112907/c-182555

http://www.summitpost.org/petit-dru/108236/c-150757

http://www.summitpost.org/petit-dru/108291/c-150757

http://www.summitpost.org/les-drus-by-sjaak-de-visser/108214/c-150757

*Camptocamp.org (3)*

http://s.camptocamp.org/uploads/images/1059260673_1126855737.jpg

http://s.camptocamp.org/uploads/images/1002626915_1423457826.jpg

http://s.camptocamp.org/uploads/images/1286183275_392583017.jpg

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

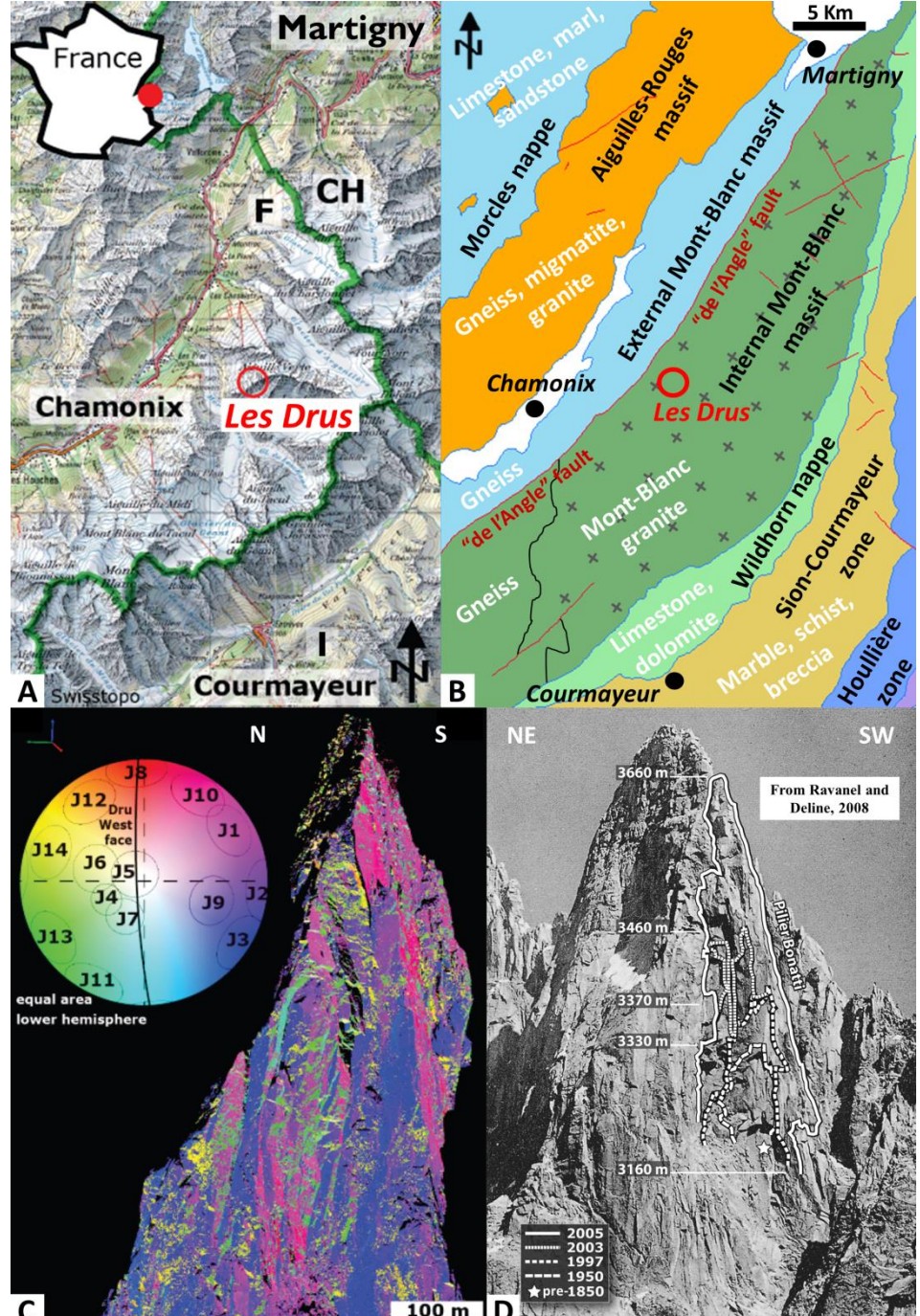

**Figure 1: Location and geological setting of the study area. A: Location of the Drus Mountain within the Chamonix Valley (Mont-Blanc massif, France); background map: Swisstopo. B: Geotectonic map of the study area (map modified after Steck et al., 2000). C: 2005/2011 merged LiDAR point cloud and discontinuities measured in the Drus West face. Each color corresponds to the stereographic projection of the poles of joint sets (Schmidt stereonet, Coltop3D software). D: Photo-comparison reconstruction of the main historical rockfall events occurred in the Drus West face since 1850 (figure modified after Ravanel and Deline, 2008).**

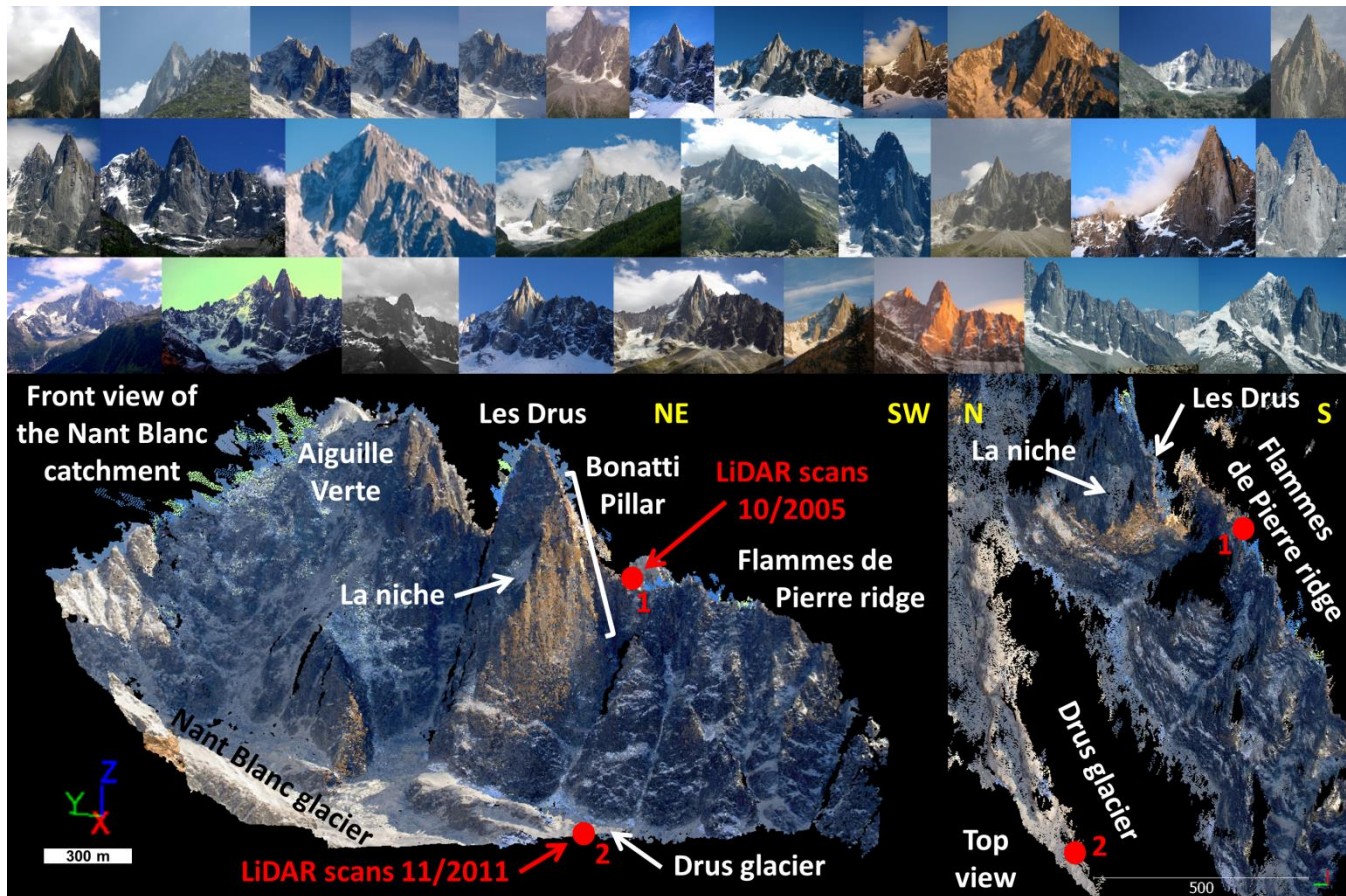

**Figure 2: Catalogue of the 30 pictures selected on the Internet (upper part, links available in the Appendix A) and used to reconstruct the north-western side (the Nant Blanc catchment) of the Aiguille Verte and Drus (lower part, front view and top view of the SfM point cloud) before the Bonatti Pillar collapse in June 2005. Both red dots show the location of the 2005 and 2011 ground-based LiDAR acquisitions.**

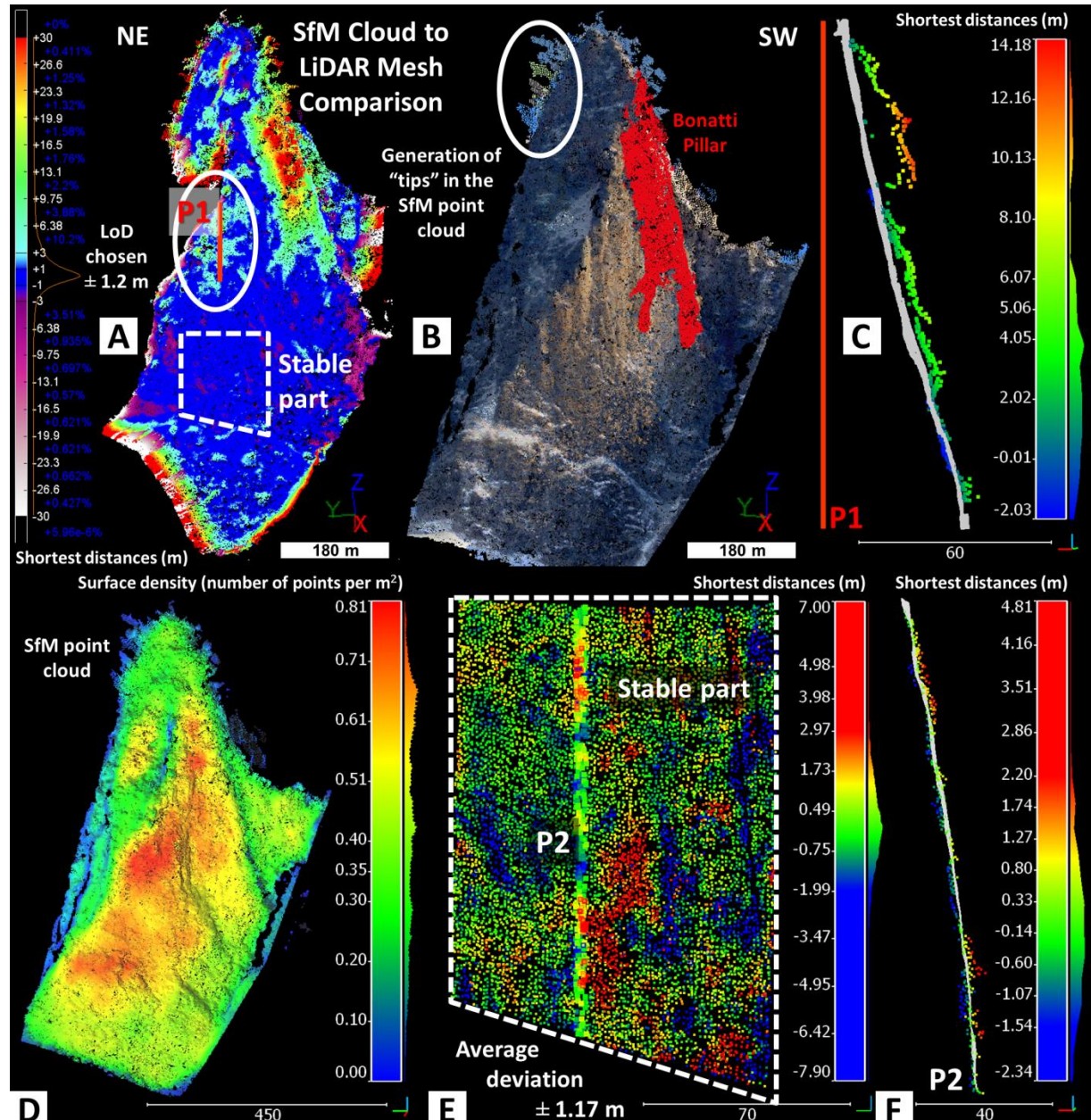

**Figure 3: SfM/LiDAR comparison. A:** Result of the point-to-mesh comparison between the SfM point cloud and the reference LiDAR mesh. The color scale of the shortest distances is divided in two parts: positive deviations from blue to red and negative deviations from blue to white. **B:** SfM point cloud of the Drus Mountain with in red, the positive deviations extracted from the comparison and associated with the Bonatti Pillar collapse. The two white ellipses illustrate the artefacts that form "tips" in the SfM model and the red line located in the center of the left ellipse corresponds to the longitudinal cross-section P1 that passes through the LiDAR mesh and the SfM point cloud. This cross-section is visible in C where the grey points correspond to the LiDAR mesh while the colored points come from the SfM model. **D:** Point density map of the SfM model (number of points per m$^2$). **E:** Point-to-mesh deviations observed in the stable area framed in A with the localisation of the longitudinal cross-section P2 (highlighted points). This cross-section is visible in F where the grey points correspond to the LiDAR mesh while the colored points come from the SfM model.

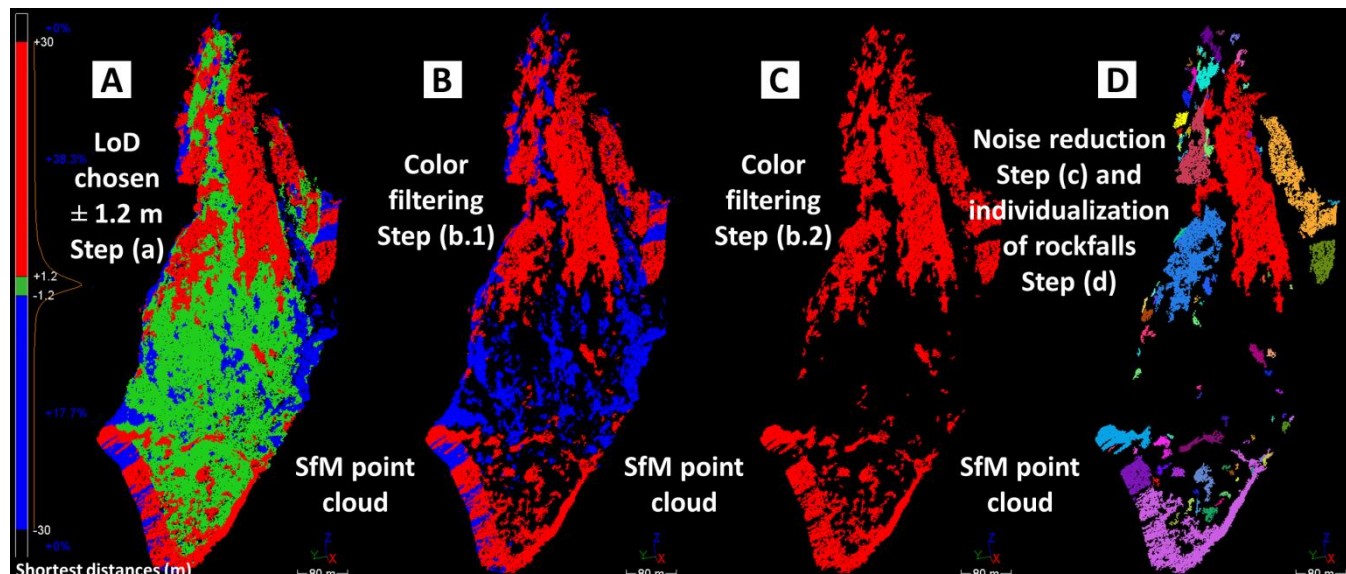

**Figure 4: Rockfall extraction method. A: Result of the point-to-mesh comparison between the SfM point cloud and the reference LiDAR mesh. Same result as in Fig. 3A but displayed with three colors and a LoD of ± 1.2 m. Red color: positive deviations, green color: points between ± 1.2 m and blue color: negative deviations. B: Filtering of the green color (first step). C: Filtering of the blue color (second step). D: Rockfall extraction and individualization. Each color corresponds to an individual point cloud. The points associated with the Bonatti Pillar collapse were highlighted in red.**

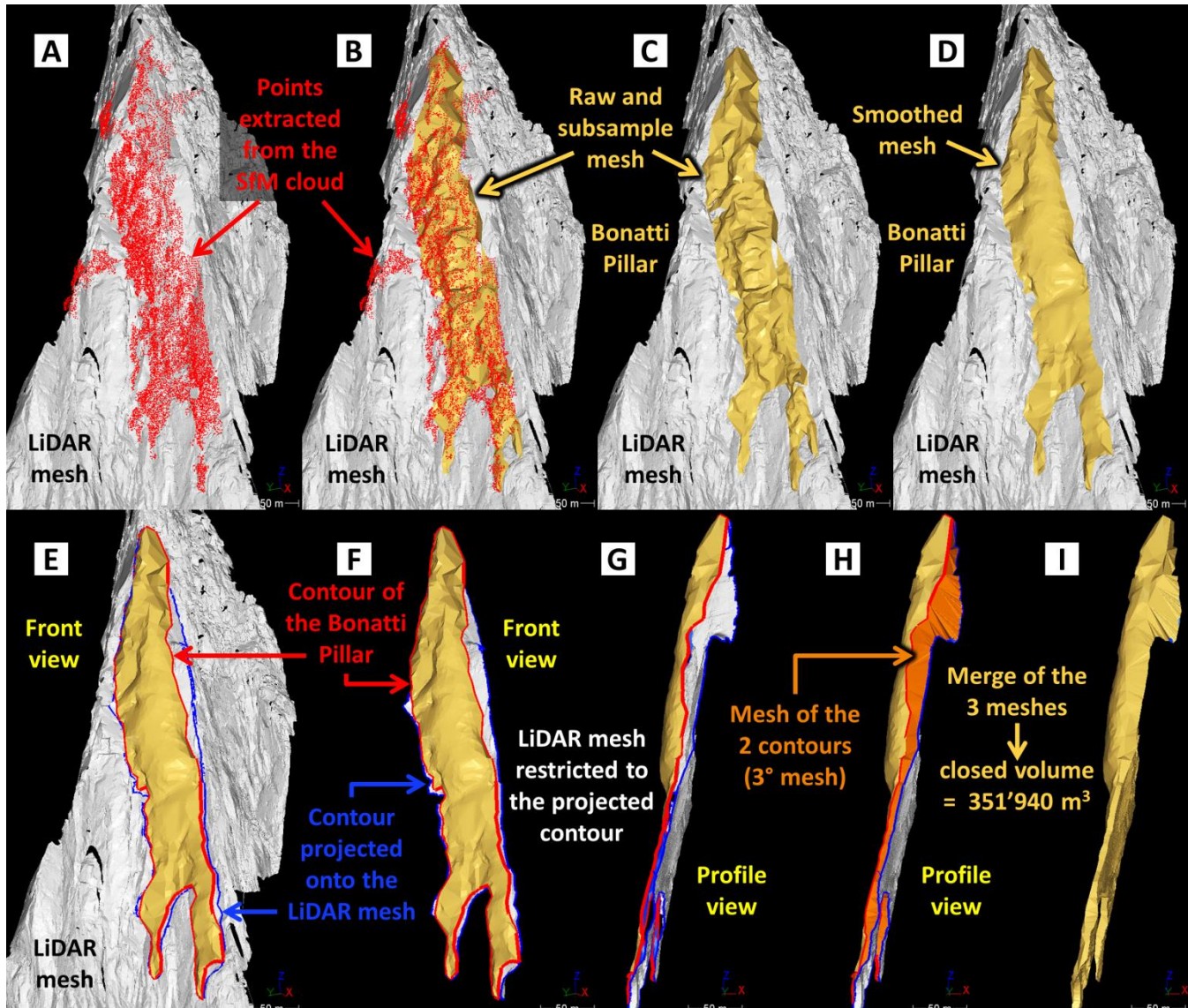

**Figure 5: Volume calculation method. A: Front view of the LiDAR mesh with the extracted SfM points highlighted in red in Fig. 4D and associated with the Bonatti Pillar collapse. B: Same image as before but with the raw and subsample SfM mesh. C: Front view of the LiDAR mesh with the raw and subsample SfM mesh. D: Front view of the LiDAR mesh with the smoothed SfM mesh. E: Same image as before but with the contour (in red) of the smoothed SfM mesh and the contour (in blue) orthogonally projected onto the LiDAR mesh. F: Front view of the smoothed SfM mesh, its contour and the projected contour that allowed cutting the LiDAR mesh. G: Profile view of the smoothed SfM mesh, its contour and the projected contour that allowed cutting the LiDAR mesh. H: Same image as before but with the third mesh (in orange) that connects the two contours. I: Profile view of the volume collapsed between June 2005 and November 2011.**

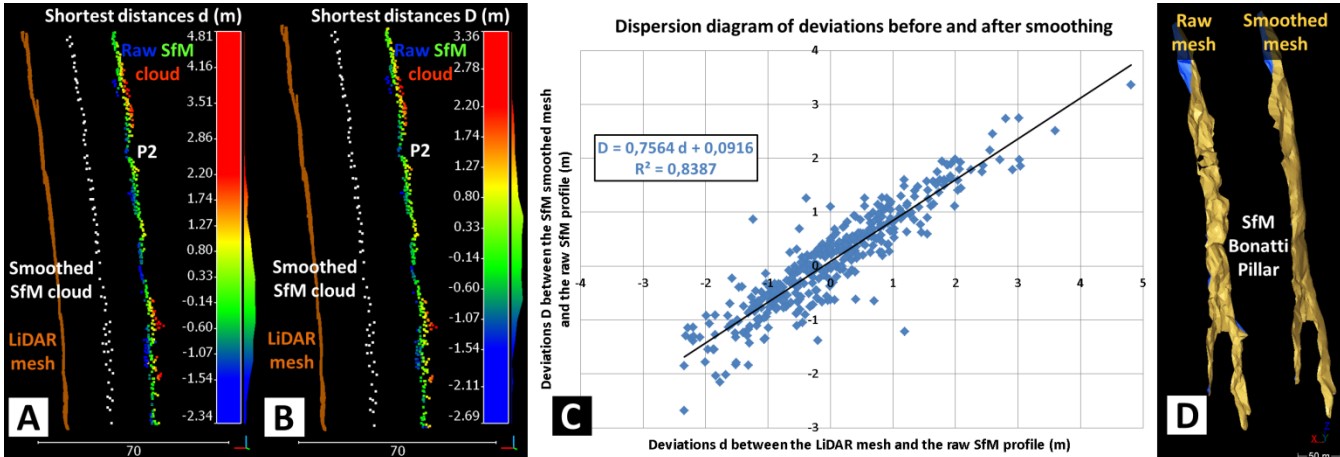

**Figure 6: Smoothed data versus raw data. A: Result of the point-to-mesh comparison between the raw SfM profile P2 (colored points) and the reference LiDAR mesh (in brown). Same result as in Fig. 3F but with the subsample and smoothed SfM profile P2 (in white). The three entities were shifted in order to better appreciate the differences between each profile. B: Result of the point-to-mesh comparison between the raw SfM profile P2 (colored points) and the subsample and smoothed SfM profile P2 (in white). C: Dispersion diagram showing the interest of smoothing in the case of the SfM point cloud: minimization of large deviations and reduction of the average deviation by ± 0.76 m. D: Profile view of the raw and smoothed SfM meshes present in Fig. 5C and 5D.**

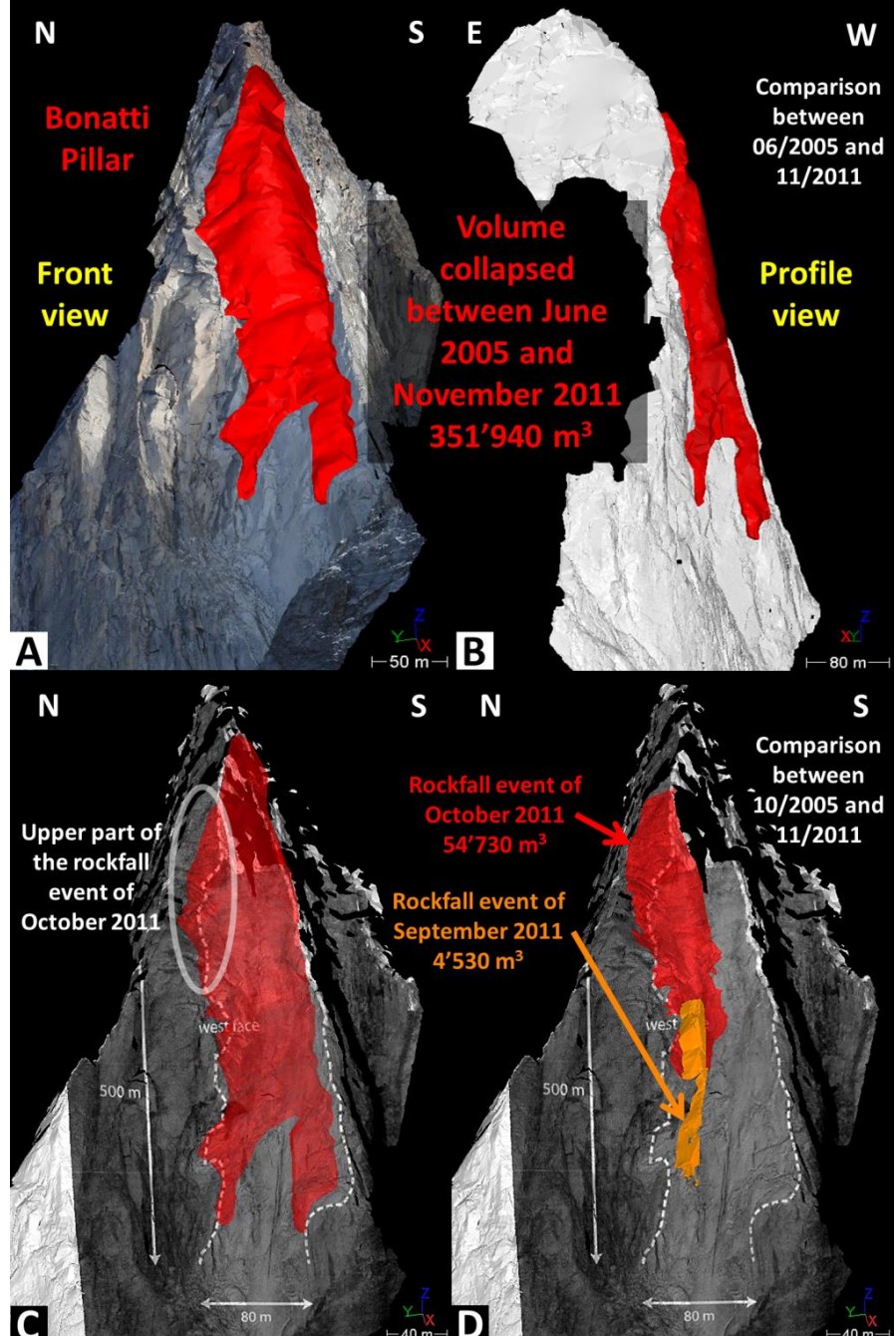

**Figure 7: 3D geometry of the volumes collapsed between June 2005 and November 2011. A: Photorealistic model of the Drus West face (high-resolution LiDAR mesh textured with a picture of November 2011) and front view of the volume (in red) calculated with 3DReshaper. B: Profile view of the high-resolution LiDAR mesh (non-textured) as well as the volume shown in Fig. 4A. C: Superimposition of the volume collapsed between June 2005 and November 2011 with another photorealistic model, textured from the left part of the Fig. 8 of Fort et al. (2009). The white dashed line shows the scar limits of the June 2005 rock-avalanche and the white ellipse illustrates the area that corresponds to the upper part of the October 2011 rockfall event. D: Superimposition of the rockfall events occurred in September (in orange) and October 2011 (in red) with the same photorealistic model as before.**