# Peer review of "Brief communication "3D reconstruction of a collapsed rock pillar from web-retrieved images and terrestrial LiDAR data – The 2005 event of the West face of the Drus (Mont-Blanc massif)""

_Natural Hazards and Earth System Sciences, 2016_

## Referee Comment (RC1) · Anonymous Referee #1 · 16 Nov 2016

The article presents an approach for quantifying the volume of rock slides when pre-slide data was not acquired, using post-slide LiDAR scans and pre-slide structure from motion (SfM) tools based on public domain photos. This approach is applicable in many other cases of mass movements, such as coastal retreat, landslides, etc. Especially in areas where there is no constant monitoring of the terrain, and yet there are enough available photos of preceding period prior to the mass movement.

The article is well written and concise. Using SfM and LiDAR is not a new approach to quantifying mass movement, but the presented case study is very interesting and

provides a chance to examine the pros and cons of this approach. However the article is lacking discussion on the various possible sources of errors and how to quantify them. In page 5, row 27, it is mentioned that there is a 5% error on the determined volume, but this is only derived from the limited resolution of the SfM method itself. Other error sources are mentioned but not always quantified.

The range of relative error mentioned in page 6, row 5 is calculated as a percentage from the "overestimated" volume when it should be calculated from the comparison volume, so that the percentages should be 10.4% and 23.1% instead of 9% and 19%.

The paper could benefit from a separate discussion about error sources and how much they affect the final result. The right part of figure 3 is a good start – it shows that even in areas where there was no known mass movement, there is still a difference between SfM and LiDAR. This could be used for estimating error per area of scan.

Another point is the volume calculation section – this is one of the thornier problems in many monitoring studies, how to estimate change in 3D volumes. The whole section is somewhat cryptic to me, and while I understand the need to keep the text short, there are no references to a detailed description of the method in the whole section. Did the authors use built-in functionality in 3Dreshaper? I know that many readers will be interested in that particular part of the article so it would be beneficial to expand upon how the volumes are extracted and subtracted, maybe with an accompanying illustration.

Finally, some nitpicking:

Page 1, row 27: "legendary climbing routes" is a term for "basecamp", not for NHESS

Page2, row 12: city of Kathmandu, not Kathmandu city.

Page 3, row 22: when you mention neglecting the snow, do you ignore it completely or mask the snowy parts from the image? And if you ignore it, does it not affect the final image?

Page 4, row 11: Mean density is not always a useful metric, especially if point density is very variable. Please specify the resolution of the final model, the standard deviation or add a point density map. If there are low density zones in critical areas that could affect the final result and become a significant source of error.

Page 5, row 13: by relative error I assume you mean between your 3 SfM models?

Page 5, row 22: what is normal about the difference? Do you mean it is expected?

Page 6, row 29: who are the Bisson brothers?

Figure 1b: The yellow unit is undefined.

Figure 1d: should be "pre-1850", not "avant"

Figure 3: should have a, b and c for easier referencing.

Figure 4d: please add the dashed scar limit so the comparison with 4c will be easier

———————————————

---

## Referee Comment (RC2) · Anonymous Referee #2 · 16 Mar 2017

The manuscript by Guerin et al. describes an interesting and potentially important possibility to reuse historical images for landscape reconstruction. In this study the volume of a collapsed rock pillar is reconstructed using TLS data for the post-event and using historical photos to perform image-based surface reconstruction of the pre-event. The introduced method should be very relevant for many geo-scientific applications when aspects of landscape evolution are of interest. Already Bakker & Lane (2016) showed the potential using archives of aerial images. Expanding it to terrestrial cases is another great opportunity.

The manuscript is well structured. However, there are some issues regarding accuracy assessment, especially considering the SfM point cloud from historical images. The study shows that it is possible to reconstruct the surface but due to missing GCP implementation and long distances between camera and object well-grounded accuracy estimation would be desirable to reveal if the reconstruction is also veritable. To me, the used reference of a former study seems to be critical because there are no statements regarding its reliability. If this issue is accounted for the paper would make a great contribution in the field of natural hazard investigations.

—-

Detailed comments:

Page 1 line 29/30: Please, be more specific about the approach to estimate rock thickness from historical images without doing 3D reconstruction. How reliable is it? This is also relevant because you will compare your own results to this study.

Page 2 line 14 -21: Maybe, also refer to Eltner et al. (2016) because the authors give a review on SfM used in geosciences and furthermore summarise accuracies achieved at different scales. As well, Smith et al. (2015) could be cited as they review applications and explain the workflow.

Page 3 line 27: If you merge scans from 2005 to 2010 to achieve a detailed 3D model of the upper face, how certain are you that no changes occurred between 2005 and 2010 to allow for a reliable model?

Page 4 line 2: Could you geo-reference with ICP because the source cloud for alignment was already geo-located?

Page 4 line 3: What do you mean by accurate GPS? Do you refer to dGPS? Furthermore, what did you measure with GPS? The scan position or marker positions?

Page 4 line 6: subtracted

Page 4 line 14/15: Did you identify stable areas and subsequently use these to perform ICP? If not, how did you account for potential errors in this regard (e.g. see Wujanz et al., 2016)

Page 4 line 22-26: I am afraid, I did not understand the procedure. Is the point extraction performed with the point cloud containing the cloud-to-mesh information? Thus, does it account on point cloud distances or solely the topographic information of a single cloud?

Page 4 line 24: How did you define the LoD (what is your accuracy measure)? Chapter 2.4: What is the average deviation between SfM and Lidar in the stable areas? This could be helpful information to better assess the performance of SfM. Furthermore, the accuracy would be interesting because many images seem to be taken from similar perspectives leading to an unfavourable base-height-ratio potentially resulting in lower accuracies.

Page 5 line 12/13: Why do you choose this as average volume? Furthermore, the random selection of image number (84% and 67%) does not seem to be sufficient to allow for the statement of a relative error. This also accounts to page 6 line 5-7, when the Ravanel & Deline (2008) value is chosen as reference.

—-

References:

Bakker, M., Lane, S. (2016). Archival photogrammetric analysis of river-floodplain systems using Structure from Motion (SfM) methods: Archival photogrammetric analysis of river systems using SfM methods. Earth Surface Processes and Landforms.

Eltner, A., Kaiser, A., Castillo, C., Rock, G., Neugirg, F., & Abellan, A. (2016). Image-based surface reconstruction in geomorphometry – merits, limits and developments. Earth Surface Dynamics, 4, 359–389.

Smith, M. W., Carrivick, J. L., & Quincey, D. J. (2015). Structure from motion photogrammetry in physical geography. Progress in Physical Geography, 1–29.

Wujanz, D., Krueger, D., Neitzel, F. (2016): Identification of Stable Areas in Unreferenced Laser Scans for Deformation Measurement. The Photogrammetric Record, 31(155), 261-280

---

## Author Comment (AC1) · 12 May 2017

Response letter to the Anonymous Referee #1 of the NHESS journal – Guerin et al.: "3D reconstruction of a collapsed rock pillar from web-retrieved images and terrestrial LiDAR data – The 2005 event of the West face of the Drus (Mont-Blanc massif)"

The authors took into account all the general comments and required corrections by both reviewers in the new version of this Brief communication which has been well developed. We first try to quantify more precisely the different sources of errors that influence the estimation of the final volume and to relate them to each step of the method

(number of photos taken into account, SfM method itself, scaling/alignment procedure and mesh smoothing procedure). The authors have also developed the section 2 "Material and methods" in order to clarify the different stages of data processing. Then, on the basis of the Fig. 3C, several illustrations have been added (Fig. 3E and 3F, Fig. 6) in order to better characterize the deviations resulting from the point-to-mesh comparison, particularly in stable areas. Furthermore, as proposed by the reviewer #1, a point density map was added (Fig. 3D) as well as Fig. 4 and Fig. 5, so as to better illustrate the methods of rockfall extraction and volume calculation. Finally, the four references suggested by the reviewer #2 have been incorporated into the manuscript. In italics below, you will find the answers to the questions you have asked.

Reviewer #1's questions

Page 1, row 27: "legendary climbing routes" is a term for "basecamp", not for NHESS. We have removed the word "legendary" from this sentence (page 3, row 27).

Page2, row 12: city of Kathmandu, not Kathmandu city. This modification was taken into account in page 4 (row 14).

Page 3, row 22: when you mention neglecting the snow, do you ignore it completely or mask the snowy parts from the image? And if you ignore it, does it not affect the final image? Indeed, it was not very clear... For clarity, we replaced "However, snow is hardly present in the steep Drus faces and its influence can be neglected on the winter images." by "However, in winter, snow is hardly present in the steep Drus faces and except at the foot of the cliff, there is no snow in the area of interest of the Bonatti Pillar on the 30 selected images." (page 6, row 1-3)

Page 4, row 11: Mean density is not always a useful metric, especially if point density is very variable. Please specify the resolution of the final model, the standard deviation or add a point density map. If there are low density zones in critical areas that could affect the final result and become a significant source of error. As you suggested above, a point density map has been added (Fig. 3D) and the mean density of points per m2 in

the area of interest of the Bonatti Pillar was specified in section 2.4 (page 7, row1).

Page 5, row 13: by relative error I assume you mean between your 3 SfM models? Yes in this paragraph, we are talking about the relative error between the 3 SfM models. This modification was taken into account in page 8 (row 32).

Page 5, row 22: what is normal about the difference? Do you mean it is expected? Yes, we wanted to say "expected". This modification was taken into account in page 9 (row 27).

Page 6, row 29: who are the Bisson brothers? The Bisson brothers are two pioneers of the French photography. This information was added at the end of the conclusion (page 10, row 21).

Figure 1b: The yellow unit is undefined. The geotectonic map of Fig. 1B (as well as the related references) has been modified following the advice of M. Jean-Luc Epard, Professor of Structural Geology at the University of Lausanne. All units are now well defined (page 15).

Figure 1d: should be "pre-1850", not "avant". This modification was taken into account in Fig. 1D (page 15).

Figure 3: should have a, b and c for easier referencing. This modification was taken into account in Fig. 3 (page 17).

Figure 4d: please add the dashed scar limit so the comparison with 4c will be easier. This modification was taken into account in Fig. 7D (page 21)

Please also note the supplement to this comment:
http://www.nat-hazards-earth-syst-sci-discuss.net/nhess-2016-316/nhess-2016-316-AC1-supplement.pdf

**Supplement:**

**Response letter to the Anonymous Referee #1 of the NHESS journal – Guerin et al.: "3D reconstruction of a collapsed rock pillar from web-retrieved images and terrestrial LiDAR data – The 2005 event of the West face of the Drus (Mont-Blanc massif)"**

The authors took into account all the general comments and required corrections by both reviewers in the new version of this Brief communication which has been well developed. We first try to quantify more precisely the different sources of errors that influence the estimation of the final volume and to relate them to each step of the method (number of photos taken into account, SfM method itself, scaling/alignment procedure and mesh smoothing procedure). The authors have also developed

10 the section 2 "Material and methods" in order to clarify the different stages of data processing. Then, on the basis of the Fig. 3C, several illustrations have been added (Fig. 3E and 3F, Fig. 6) in order to better characterize the deviations resulting from the point-to-mesh comparison, particularly in stable areas. Furthermore, as proposed by the reviewer #1, a point density map was added (Fig. 3D) as well as Fig. 4 and Fig. 5, so as to better illustrate the methods of rockfall extraction and volume calculation. Finally, the four references suggested by the reviewer #2 have been incorporated into the manuscript. In italics

15 below, you will find the answers to the questions you have asked.

**Reviewer #1's questions**

Page 1, row 27: "legendary climbing routes" is a term for "basecamp", not for NHESS.

20 *We have removed the word "legendary" from this sentence (page 3, row 27).*

Page2, row 12: city of Kathmandu, not Kathmandu city.

*This modification was taken into account in page 4 (row 14).*

25 Page 3, row 22: when you mention neglecting the snow, do you ignore it completely or mask the snowy parts from the image? And if you ignore it, does it not affect the final image?

*Indeed, it was not very clear… For clarity, we replaced "However, snow is hardly present in the steep Drus faces and its influence can be neglected on the winter images." by "However, in winter, snow is hardly present in the steep Drus faces and except at the foot of the cliff, there is no snow in the area of interest of the Bonatti Pillar on the 30 selected images."*

30 *(page 6, row 1-3)*

Page 4, row 11: Mean density is not always a useful metric, especially if point density is very variable. Please specify the resolution of the final model, the standard deviation or add a point density map. If there are low density zones in critical areas that could affect the final result and become a significant source of error.

*As you suggested above, a point density map has been added (Fig. 3D) and the mean density of points per $m^2$ in the area of*

5  *interest of the Bonatti Pillar was specified in section 2.4 (page 7, row1).*

Page 5, row 13: by relative error I assume you mean between your 3 SfM models?

*Yes in this paragraph, we are talking about the relative error between the 3 SfM models.*

*This modification was taken into account in page 8 (row 32).*

Page 5, row 22: what is normal about the difference? Do you mean it is expected?

*Yes, we wanted to say "expected". This modification was taken into account in page 9 (row 27).*

Page 6, row 29: who are the Bisson brothers?

15  *The Bisson brothers are two pioneers of the French photography.*

*This information was added at the end of the conclusion (page 10, row 21).*

Figure 1b: The yellow unit is undefined.

*The geotectonic map of Fig. 1B (as well as the related references) has been modified following the advice of M. Jean-Luc*

20  *Epard, Professor of Structural Geology at the University of Lausanne. All units are now well defined (page 15).*

Figure 1d: should be "pre-1850", not "avant".

*This modification was taken into account in Fig. 1D (page 15).*

25  Figure 3: should have a, b and c for easier referencing.

*This modification was taken into account in Fig. 3 (page 17).*

Figure 4d: please add the dashed scar limit so the comparison with 4c will be easier.

*This modification was taken into account in Fig. 7D (page 21).*

[revised manuscript text omitted]

---

## Author Comment (AC2) · 12 May 2017

Response letter to the Anonymous Referee #2 of the NHESS journal – Guerin et al.: "3D reconstruction of a collapsed rock pillar from web-retrieved images and terrestrial LiDAR data – The 2005 event of the West face of the Drus (Mont-Blanc massif)"

The authors took into account all the general comments and required corrections by both reviewers in the new version of this Brief communication which has been well developed. We first try to quantify more precisely the different sources of errors that influence the estimation of the final volume and to relate them to each step of the method

(number of photos taken into account, SfM method itself, scaling/alignment procedure and mesh smoothing procedure). The authors have also developed the section 2 "Material and methods" in order to clarify the different stages of data processing. Then, on the basis of the Fig. 3C, several illustrations have been added (Fig. 3E and 3F, Fig. 6) in order to better characterize the deviations resulting from the point-to-mesh comparison, particularly in stable areas. Furthermore, as proposed by the reviewer #1, a point density map was added (Fig. 3D) as well as Fig. 4 and Fig. 5, so as to better illustrate the methods of rockfall extraction and volume calculation. Finally, the four references suggested by the reviewer #2 have been incorporated into the manuscript. In italics below, you will find the answers to the questions you have asked.

Reviewer #2's questions

Page 1 line 29/30: Please, be more specific about the approach to estimate rock thickness from historical images without doing 3D reconstruction. How reliable is it? This is also relevant because you will compare your own results to this study. Indeed, it was not very clear... For clarity, we replaced "The assessment of this volume by Ravanel and Deline (2008) was performed in two steps: (a) determination of the rock-avalanche scar dimensions (height and width) by making measurements on terrestrial LiDAR data acquired just after the event (October 2005); and (b) estimation of the thickness of the fallen blocks from historical photographs taken from different viewpoints" by "The assessment of this volume by Ravanel and Deline (2008) was performed in two steps: (a) identification on photos of different rock elements (slabs, dihedrons, overhangs) whose dimensions (height, width, depth) can be compared with compartments now collapsed; and (b) measurements of these dimensions on terrestrial LiDAR scans acquired just after the event in October 2005. Historical photographs of the West face taken from different viewpoints facilitate the estimation of the thickness of the missing elements, which remains the most difficult dimension to determine. Under this method, the assessment of rock thickness (8 meters on average) represents the greatest source of uncertainty since the height and width of the rock-avalanche scar could be very accurately measured on the October 2005 LiDAR data." (page 4, line 27 to page 5, line 3)

Page 2 line 14 -21: Maybe, also refer to Eltner et al. (2016) because the authors give a review on SfM used in geosciences and furthermore summarise accuracies achieved at different scales. As well, Smith et al. (2015) could be cited as they review applications and explain the workflow. Eltner et al. (2016) as well as Smith et al. (2016) have been cited in the introduction (page 5, line 19 and page 5, line 24-25). Smith et al. was then quoted for the SfM workflow at the beginning of section 2.4 (page 7, line 28).

Page 3 line 27: If you merge scans from 2005 to 2010 to achieve a detailed 3D model of the upper face, how certain are you that no changes occurred between 2005 and 2010 to allow for a reliable model? Actually, we did not merge the scans from 2005 to 2010. We just used and merged the point clouds of October 2005 and November 2011 after deleting the points belonging to the collapses of September and October 2011, which include the small rockfalls detected by Ravanel (2010) between October 2005 and September 2008. For more information, please refer to section 2.3 (page 7, line 17-26).

Page 4 line 2: Could you geo-reference with ICP because the source cloud for alignment was already geo-located? Yes, "in the absence of a fairly accurate DEM (the resolution of the IGN's DEM is only 30 m in this sector), both datasets were georeferenced using the scanner position measured by dGPS, then aligned with respect to the vertical axis using the coordinates of several points distributed in the cliff and measured with a total station." (page 7, line 15-17)

Page 4 line 3: What do you mean by accurate GPS? Do you refer to dGPS? Furthermore, what did you measure with GPS? The scan position or marker positions? Please refer to the answer to the previous comment.

Page 4 line 6: subtracted. This modification was taken into account in page 7 (line 24).

[Figure]

Page 4 line 14/15: Did you identify stable areas and subsequently use these to perform ICP? If not, how did you account for potential errors in this regard (e.g. see Wujanz et al., 2016). All alignments performed using the ICP algorithms have been applied to the stable areas, whether to align both LiDAR point clouds with each other or to align by parts the SfM model on the 2005/2011 merged and cleaned LiDAR point cloud. This information has been added in sections 2.3/2.4 and Wujanz et al. (2016) has been cited in section 2.4 (page 8, line 5).

Page 4 line 22-26: I am afraid, I did not understand the procedure. Is the point extraction performed with the point cloud containing the cloud-to-mesh information? Thus, does it account on point cloud distances or solely the topographic information of a single cloud? Sections 2.5 (SfM/LiDAR comparison and rockfall extraction) and 2.6 (Volume calculation) have been substantially modified and Fig. 4, 5 and 6 (pages 19, 20 and 21) have been added to improve the understanding of the methods used in these two paragraphs. I hope it's clearer now...

Page 4 line 24: How did you define the LoD (what is your accuracy measure)? Chapter 2.4: What is the average deviation between SfM and Lidar in the stable areas? This could be helpful information to better assess the performance of SfM. Furthermore, the accuracy would be interesting because many images seem to be taken from similar perspectives leading to an unfavourable base-height-ratio potentially resulting in lower accuracies. The LoD ($\pm$ 1.2 m in our case) has been defined "in agreement with the average deviation observed in the stable areas" (page 8, line 21-22) which "reaches $\pm$ 1.17 m" (page 8, line 10). This value was determined from the result of the point-to-mesh comparison and refers to the area framed in Fig. 3A, whose the zoom is visible in Fig. 3E (page 18). Furthermore, in order to better appreciate the reconstruction of the depth in the SfM model, a top view has been added in Fig. 2 (page 17).

Page 5 line 12/13: Why do you choose this as average volume? Furthermore, the random selection of image number (84% and 67%) does not seem to be sufficient to allow for the statement of a relative error. This also accounts to page 6 line 5-7, when

the Ravanel & Deline (2008) value is chosen as reference. Indeed, choose 311'970 m3 as average volume was not very judicious... Instead, "we consider the volume of 292'680 m3 as the most reliable estimation" (page 9, line 31-32) to define our relative error range. However, in the other error calculation you mentioned, we kept the Ravanel and Deline (2008) value as reference "because of its lower uncertainty range: $\pm$ 3.8 %)". (page 10, line 33)

Please also note the supplement to this comment:
http://www.nat-hazards-earth-syst-sci-discuss.net/nhess-2016-316/nhess-2016-316-AC2-supplement.pdf
* * *
[Figure]

**Supplement:**

**Response letter to the Anonymous Referee #2 of the NHESS journal – Guerin et al.: "3D reconstruction of a collapsed rock pillar from web-retrieved images and terrestrial LiDAR data – The 2005 event of the West face of the Drus (Mont-Blanc massif)"**

The authors took into account all the general comments and required corrections by both reviewers in the new version of this Brief communication which has been well developed. We first try to quantify more precisely the different sources of errors that influence the estimation of the final volume and to relate them to each step of the method (number of photos taken into account, SfM method itself, scaling/alignment procedure and mesh smoothing procedure). The authors have also developed

10   the section 2 "Material and methods" in order to clarify the different stages of data processing. Then, on the basis of the Fig. 3C, several illustrations have been added (Fig. 3E and 3F, Fig. 6) in order to better characterize the deviations resulting from the point-to-mesh comparison, particularly in stable areas. Furthermore, as proposed by the reviewer #1, a point density map was added (Fig. 3D) as well as Fig. 4 and Fig. 5, so as to better illustrate the methods of rockfall extraction and volume calculation. Finally, the four references suggested by the reviewer #2 have been incorporated into the manuscript. In italics

15   below, you will find the answers to the questions you have asked.

**Reviewer #2's questions**

Page 1 line 29/30: Please, be more specific about the approach to estimate rock thickness from historical images without

20   doing 3D reconstruction. How reliable is it? This is also relevant because you will compare your own results to this study.

*Indeed, it was not very clear… For clarity, we replaced "The assessment of this volume by Ravanel and Deline (2008) was performed in two steps: (a) determination of the rock-avalanche scar dimensions (height and width) by making measurements on terrestrial LiDAR data acquired just after the event (October 2005); and (b) estimation of the thickness of the fallen blocks from historical photographs taken from different viewpoints" by "The assessment of this volume by Ravanel*

25   *and Deline (2008) was performed in two steps: (a) identification on photos of different rock elements (slabs, dihedrons, overhangs) whose dimensions (height, width, depth) can be compared with compartments now collapsed; and (b) measurements of these dimensions on terrestrial LiDAR scans acquired just after the event in October 2005. Historical photographs of the West face taken from different viewpoints facilitate the estimation of the thickness of the missing elements, which remains the most difficult dimension to determine. Under this method, the assessment of rock thickness (8*

30   *meters on average) represents the greatest source of uncertainty since the height and width of the rock-avalanche scar could be very accurately measured on the October 2005 LiDAR data." (page 4, line 27 to page 5, line 3)*

Page 2 line 14 -21: Maybe, also refer to Eltner et al. (2016) because the authors give a review on SfM used in geosciences and furthermore summarise accuracies achieved at different scales. As well, Smith et al. (2015) could be cited as they review applications and explain the workflow.

5  *Eltner et al. (2016) as well as Smith et al. (2016) have been cited in the introduction (page 5, line 19 and page 5, line 24-25). Smith et al. was then quoted for the SfM workflow at the beginning of section 2.4 (page 7, line 28).*

Page 3 line 27: If you merge scans from 2005 to 2010 to achieve a detailed 3D model of the upper face, how certain are you that no changes occurred between 2005 and 2010 to allow for a reliable model?

10  *Actually, we did not merge the scans from 2005 to 2010. We just used and merged the point clouds of October 2005 and November 2011 after deleting the points belonging to the collapses of September and October 2011, which include the small rockfalls detected by Ravanel (2010) between October 2005 and September 2008. For more information, please refer to section 2.3 (page 7, line 17-26).*

15  Page 4 line 2: Could you geo-reference with ICP because the source cloud for alignment was already geo-located?
*Yes, "in the absence of a fairly accurate DEM (the resolution of the IGN's DEM is only 30 m in this sector), both datasets were georeferenced using the scanner position measured by dGPS, then aligned with respect to the vertical axis using the coordinates of several points distributed in the cliff and measured with a total station." (page 7, line 15-17)*

20  Page 4 line 3: What do you mean by accurate GPS? Do you refer to dGPS? Furthermore, what did you measure with GPS? The scan position or marker positions?
*Please refer to the answer to the previous comment.*

Page 4 line 6: subtracted.
25  *This modification was taken into account in page 7 (line 24).*

Page 4 line 14/15: Did you identify stable areas and subsequently use these to perform ICP? If not, how did you account for potential errors in this regard (e.g. see Wujanz et al., 2016).
*All alignments performed using the ICP algorithms have been applied to the stable areas, whether to align both LiDAR point*
30  *clouds with each other or to align by parts the SfM model on the 2005/2011 merged and cleaned LiDAR point cloud. This information has been added in sections 2.3/2.4 and Wujanz et al. (2016) has been cited in section 2.4 (page 8, line 5).*

Page 4 line 22-26: I am afraid, I did not understand the procedure. Is the point extraction performed with the point cloud containing the cloud-to-mesh information? Thus, does it account on point cloud distances or solely the topographic information of a single cloud?

*Sections 2.5 (SfM/LiDAR comparison and rockfall extraction) and 2.6 (Volume calculation) have been substantially modified and Fig. 4, 5 and 6 (pages 19, 20 and 21) have been added to improve the understanding of the methods used in these two paragraphs. I hope it's clearer now…*

Page 4 line 24: How did you define the LoD (what is your accuracy measure)? Chapter 2.4: What is the average deviation between SfM and Lidar in the stable areas? This could be helpful information to better assess the performance of SfM. Furthermore, the accuracy would be interesting because many images seem to be taken from similar perspectives leading to an unfavourable base-height-ratio potentially resulting in lower accuracies.

*The LoD (± 1.2 m in our case) has been defined "in agreement with the average deviation observed in the stable areas" (page 8, line 21-22) which "reaches ± 1.17 m" (page 8, line 10). This value was determined from the result of the point-to-mesh comparison and refers to the area framed in Fig. 3A, whose the zoom is visible in Fig. 3E (page 18). Furthermore, in order to better appreciate the reconstruction of the depth in the SfM model, a top view has been added in Fig. 2 (page 17).*

Page 5 line 12/13: Why do you choose this as average volume? Furthermore, the random selection of image number (84% and 67%) does not seem to be sufficient to allow for the statement of a relative error. This also accounts to page 6 line 5-7, when the Ravanel & Deline (2008) value is chosen as reference.

*Indeed, choose 311'970 m$^3$ as average volume was not very judicious… Instead, "we consider the volume of 292'680 m$^3$ as the most reliable estimation" (page 9, line 31-32) to define our relative error range. However, in the other error calculation you mentioned, we kept the Ravanel and Deline (2008) value as reference "because of its lower uncertainty range: ± 3.8 %)". (page 10, line 33)*

[revised manuscript text omitted]